# Differences in Oral Health and Generalized Anxiety Disorder According to Secondhand Smoke Exposure in Public Places

**DOI:** 10.3390/bs13060455

**Published:** 2023-05-31

**Authors:** Yu-Rin Kim, Kyeung-Ae Jang

**Affiliations:** Department of Dental Hygiene, Silla University, 140 Baegyang-daero 700 Beon-gil, Sasang-gu, Busan 46958, Republic of Korea; dbfls1712@silla.ac.kr

**Keywords:** anxiety disorders, Korea, oral health, smoke

## Abstract

Background: This study aims to provide basic data for minimizing exposure to secondhand smoke (SHS) by confirming the relationship between exposure to SHS and generalized anxiety disorder (GAD). Methods: Using the third year data of the eighth National Health and Nutrition Examination Survey, 3874 people were selected as subjects. Complex sampling analysis was applied to all analyses, and 307 people were exposed to the SHS group (SHSG) and 3567 people were not exposed to the SHS group (NSHSG). Complex sample linear regression analysis was conducted to confirm the effect of exposure to SHS on oral health and GAD. Results: Among oral-health-related factors, exposure to SHS in Korean adults was related to the presence of implants. Additionally, exposure to SHS had a significant effect on GAD despite adjusting for sociodemographic characteristics and oral-health-related factors (*p* < 0.05). Conclusions: In this study, the relationship between passive smoking and GAD was confirmed. Therefore, to reduce GAD, oral health management is necessary and exposure to SHS should be minimized.

## 1. Introduction

According to the Korea National Health and Nutrition Examination Survey (KNHNES) of 2021, the smoking rate among Korean adults is 19.3%, more specifically 31.3% for men and 6.9% for women [1]. Standards for smoking rate statistics are different in Korea and abroad. In Korea, “daily smokers” and “social (occasional) smokers” are added together for adults aged 19 years or older, but, internationally, only “daily smokers” are counted for the population aged 15 years or older. According to international standards, the smoking rate in Korea is 15.9% as of 2020, which is similar to the average smoking rate in OECD countries [1]. In the past, smoking was regarded as an individual’s preference, but now it is defined as a major risk factor that harms not only one’s own health but also the health of people around them through secondhand smoke (SHS) [2,3].

SHS generally refers to indirectly inhaling smoke generated by burning cigarettes or cigarettes smoked by others [4]. This is known to adversely affect the lungs and the vascular system, especially in adults, as secondhand smoking has been associated with high mortality rates, due to it being a primary cause of vascular diseases; therefore, a policy that can reduce exposure to secondhand smoking is necessary [5]. Many countries have established policies to prevent exposure to secondhand smoking, and Korea has also expanded various policies to reduce secondhand smoking exposure since the enactment of the National Health Promotion Act (1995) [6]. The study of Joaquin Barnoya and Ana Navas-Acien specified that Article 8 of the Act states that the law “protects citizens from exposure to smoke in workplaces, public transportation and public places”, but still most of the world’s population is exposed to secondhand smoking [7]. In Korea, the exposure to secondhand smoking in adolescents was 20.0% in school, 23.0% at home, and 51.4% in public places. Secondhand smoking affects oral health, which has been reported in previous studies [8,9]. Yuko Inoue et al. [8] reported that the risk of tooth loss increases when there is exposure to secondhand smoking at home, and the study among the elderly in Japan [9] also reported the connection between secondhand smoking and tooth loss. In addition, secondhand smoking is also related to mental health because it is recognized as a damage received from others. Kim and Kim [10] reported that secondhand smoking exposure is related to lack of sleep, depression, and death. Lee et al. [11] also said that secondhand smoking is relevant to mental health conditions, such as depression, stress, and suicide. One percent of the total mortality rate is likely to be associated with SHS every year, and it is very important to limit secondhand smoking to improve public health [9].

In the context of mental health, anxiety disorder refers to a mental illness that causes problems in everyday living due to various types of inappropriate and pathological anxiety. Among the types of anxiety disorders, generalized anxiety disorder (GAD) has the main symptoms of constant worry and anger, even in trivial and everyday matters, and continuous anxiety that is difficult to control [12]. If this GAD becomes chronic, the subjective health cognition of the patient can be lowered, and the quality of life can be reduced by systemic disorders [13]. In previous studies, research on secondhand smoking and systemic health [2,3,5], oral health [8,9], and mental health [10,11] have been reported, and Shin et al. [14] confirmed such for specific GAD, but the study is limited to adolescents and no research has yet to be focused on adults.

In this study, we use the GAD-7 screening tool to select newly surveyed adults in the 8th KNHNES in 2021 to confirm the difference between secondhand smoking exposure in public places, the degree of oral health care, and GAD. We will confirm the impact of secondhand smoking exposure on GAD to provide basic data for improving both general health and oral health of adults. Therefore, the hypotheses of this study are as follows.

**H_0_**.*There may be no difference between GAD and oral health depending on exposure to SHS*.

**H_1_**.*There may be a difference between GAD and oral health depending on exposure to SHS*.

## 2. Materials and Methods

### 2.1. Study Design

This study was conducted based on the annual KNHNES by the Korea Disease Control and Prevent Agency. In the 8th survey, there were 9682 subjects in the third year (2021), 7090 participants participated in one or more of the health survey, examination survey, and nutrition survey, and the participation rate was 73.2%. A total of 3886 people, aged 19 to 65, were extracted in accordance with the purpose of this study, and the final 3874 subjects were selected, excluding missing values for exposure to SHS in public places (Figure 1).

### 2.2. Setting

Among the 9682 subjects of the 2021 KNHNES, 3874 persons who met the inclusion criteria of the study were selected as the final analysis subjects. The National Health and Nutrition Survey is a legal survey on the health behavior of the people, the prevalence of chronic diseases, food, and nutrition ingestion, and is a government figure (approval No. 117002) based on Article 17 of the Statistics Act. The data used in this study were from the first year and third year of the 8th survey. The review of research ethics was resumed in consideration of the collection of human-derived materials and the provision of raw data to third parties and was approved by the IRB (Institutional Review Board; 2018-01-03-3C-A).

### 2.3. Study Participants

In this study, exposure to SHS in public places was confirmed as an independent variable. The variables of ‘yes’ and ‘no’ were configured for the contents of the questionnaire, ‘In the past 7 days, have you been exposed to cigarette smoked by other people in a public place (excluding designated smoking areas)?’ They were grouped into SHS exposure group (SHSG; 307 participants) in public places, and non-SHS exposure group (NSHSG; 3567 participants) in public places, if not applicable.

### 2.4. Variables

#### 2.4.1. Demographic Characteristics

Gender, age, marital status, income, and education level were confirmed using the health questionnaire of the KNHNES. Age was divided into ‘19–34 years old’, ‘35–44 years old’, ‘45–54 years old’, and ‘55–64 years old’, and marriage was divided into ‘married’ and ‘single’. The level of education was categorized into ‘elementary school graduate or less’, ‘middle school graduate’, ‘high school graduate’, and ‘college graduate or higher’, while income was classified by income quintile (household).

#### 2.4.2. Oral-Health-Related Factors

##### Oral-Health Status-Related Factors

Oral health status were divided into (a) presence of existing natural teeth, (b) maxillary prostheses, (c) mandibular prostheses, (d) maxillary anterior implants, (e) maxillary posterior implants, (f) mandibular anterior implants, and (g) mandibular molar implants.

Regarding oral health problems, (a) complaints of chewing discomfort, (b) chewing discomfort, speaking problems, and (c) self-recognized oral health status were confirmed. In the case of chewing and speaking problems, a higher score means no problem, and for oral health, a lower score means better health.

##### Oral-Health-Behavior-Related Factors

Oral health behavior was determined by whether or not the participant had brushed their teeth or not during the previous day, and were identified in 8 sub-items (before breakfast, after breakfast, before lunch, after lunch, before dinner, after dinner, after snack, and before bed). The treatment items of the oral examination for the past year were identified as 5 sub-items (gum disease treatment, simple tooth decay treatment, root canal treatment, preventive treatment, tooth extraction, or intraoral surgery).

#### 2.4.3. Definition of GAD

The GAD screening tool was newly introduced in the 3rd year (2021) of the 8th survey, and 7 questions were measured for adults. The items included ‘I feel nervous, anxious, or irritable’, ‘I can’t stop or control my worrying’, ‘I worry too much about many things’, I find it difficult to be comfortable’, ‘I am so restless that I have a hard time staying still’, ‘I am easily annoyed or easily offended’, and ‘I feel afraid as if something terrible is about to happen’. The response to a total of 7 questions is calculated through a scale of 0 to 3, ranging from ‘not disturbed at all’ to ‘disturbed almost every day’. A high score means a high level of GAD, and the GAD-7 total is the result of summing up 7 items, wherein a lower score indicates a higher level of GAD.

### 2.5. Data Sources/Measurement

The purpose of this study is to identify the differences in oral-health-related factors and GAD according to exposure to SHS. Among oral-health-related factors, oral-health-status-related factors were identified as oral health status (7 items), oral problems (2 items), and self-recognized oral health. Among oral-health-related factors, oral-health-behavior-related factors were identified in tooth brushing (8 items) and dental treatment (5 items).

### 2.6. Bias

When we analyzed the relationship between exposure to SHS and GAD, we presented Model 2, which corrected for sociodemographic characteristics, to minimize bias. In addition, Model 3, which corrected oral-health-related factors, was presented to analyze the influencing factors.

### 2.7. Study Size

This study was analyzed using the KNHNES data provided by the Korea Centers for Disease Control and Prevention; therefore, it was analyzed using data from 2021 when new variables for GAD were investigated.

### 2.8. Quantitative Variables

The GAD is a 4-point scale, with a higher score indicating a higher level of GAD. In addition, chewing and speaking problems were measured on a 5-point scale, and the lower the score, the more serious the problem, and for oral health, a lower score means better health.

### 2.9. Statistical Methods

Data analysis was performed using IBM SPSS ver. 21.0 (IBM Co., Armonk, NY, USA), and complex sampling analysis with stratification variables, cluster variables, and weights were applied in all analyses. A total of 3874 people were divided into 307 people exposed to SHS in public places and 3567 people not exposed to SHS in public places. A complex sample chi-square test was performed to compare demographic characteristics, oral-health-related factors. A complex sample independent *t*-test was performed to compare self-recognized oral health status, chewing discomfort, and speaking problems. A complex sample linear regression analysis was conducted to examine the effects of SHS exposure in public places on oral-health-related factors and GAD-7. In terms of the relationship between SNS and GAD-7, no adjustment was made in Model 1, demographic characteristics were adjusted in Model 2, and Model 3 was adjusted after adding oral-health-related factors to Model 2. ‘Don’t know’, ‘non-applicable’, and ‘missing values’ were all excluded in 8, 9, 88, and 99. The number of subjects in all tables was presented as an unweighted frequency, and the significance level of the statistical test (*p*-value) was 0.05.

## 3. Results

### 3.1. Demographic Characteristics of SHS Exposure in Public Places

Based on the results of confirming the demographic characteristics according to SHS exposure in public places of Korean adults, males were more exposed to SHS in public places than females, and females were more likely to have no exposure to SHS in public places (*p* < 0.05). In terms of age, 19–34 years old and 45–54 years old were exposed to SHS in public places, and 35–44 years old and 55–64 years old were mostly exposed to SHS in public places (*p* < 0.001). In terms of marital status, married people were more exposed to SHS in public places, and unmarried people were more likely to have no exposure to SHS in public places. In terms of income level, SHS exposure in public places was high in the ‘high’ and ‘low’ group. Regarding education level, there were many groups without SHS exposure in public places except for high school graduates (*p* < 0.001) (Table 1).

### 3.2. Differences in Oral-Health-Status-Related Factors According to SHS Exposure in Public Places

The difference in oral-health-status-related factors according to SHS exposure in public places in Korean adults was confirmed. There were many groups without SHS exposure in public places in all items including presence of existing natural teeth, maxillary prostheses, mandibular prostheses, maxillary anterior implants (*p* < 0.05), maxillary posterior implants (*p* < 0.05), and mandibular molar implants (*p* < 0.05). In terms of complaints of chewing discomfort, more people were exposed to SHS in public places, and the group with exposure to SHS in public places scored high in terms of self-recognized oral health status and in chewing discomfort and speaking problems, the group with SHS exposure in public places scored low, indicating high discomfort (Table 2).

### 3.3. Differences in Oral-Health-Behavior-Related Factors According to SHS Exposure in Public Places

Oral-health-behavior-related factors according to SHS exposure in public places in Korean adults was confirmed. For brushing time, the factors of before breakfast, before lunch (*p* < 0.05), before dinner, after snack, and before bedtime (*p* < 0.05) were higher in the group with SHS exposure in public places. The factors of after lunch (*p* < 0.05) and after dinner were higher in the group without SHS exposure in public places. For treatment items, oral examination, dental clinic use, and oral examination, gum disease treatment, simple tooth decay treatment, root canal treatment, tooth extraction, or intraoral surgery in the past year were more in the group with SHS exposure in public places. Oral examination (*p* < 0.01) and preventive treatment were more in the group without SHS exposure in public places (Table 3).

### 3.4. Differences in GAD-7 According to SHS Exposure in Public Places

GAD according to SHS exposure in public places in Korean adults was confirmed. In all items including ‘I feel nervous, anxious, or irritable’ (*p* < 0.001), ‘I can’t stop or control my worrying’ (*p* < 0.001), ‘I worry too much about many things’ (*p* < 0.001), ‘I find it difficult to be comfortable‘ (*p* < 0.01), ‘I am so restless that I have a hard time staying still’ (*p* < 0.01), ‘I am easily annoyed or easily offended’ (*p* < 0.001), and ‘I feel afraid as if something terrible is about to happen’ (*p* < 0.01), the group exposed to SHS in public places had a higher level of GAD (Table 4).

### 3.5. Relationship between SHS Exposure and GAD in Public Places

The relationship between SHS exposure and GAD in public places in Korean adults was confirmed. There was a significant effect on all seven sub-items and GAD-7 total if there was exposure to SHS in a public places (*p* < 0.001). In Model 2, which was adjusted for demographic characteristics, SHS exposure in public places also had a significant effect on all seven sub-items and GAD-7 total (*p* < 0.05). In Model 3, which was adjusted by adding oral-health-related factors to Model 2, SHS exposure public places also had a significant effect on all seven sub-items and GAD-7 total (*p* < 0.05) (Table 5).

## 4. Discussion

GAD is a disease characterized by the main symptom of persistent anxiety that occurs even over trivial and everyday matters [12]. The current lifetime prevalence rate in Korea is about 2.4% [15]. Compared to other types of mental health problems, GAD has ambiguous symptoms and may be masked by other comorbid mental diseases, so early detection is important [16]. Although many previous studies [17,18,19,20,21,22] reported the relationship between SHS and mental health, some studies reported that there was no relationship between them [23,24,25], so further research is required. Accordingly, this study compared demographic characteristics, differences in oral health status, oral health care practice, and GAD according to SHS exposure in public places for adults who participated in the 2021 Korea National Health and Nutrition Examination Survey. In addition, we confirmed the effect of exposure to SHS in public places on GAD and analyzed the effect by adjusting demographic characteristics and oral health care practice. Results showed the effect of passive smoking on the degree of GAD was 1.218 before adjustment, 1.225 after adjusting for demographic characteristics, and 1.423 after adjusting for oral health care practice. These results are similar to the results of a study by Shin et al. [14] targeting adolescents. According to their study, the risk of GAD was 1.17 times higher at home, 1.49 times higher in public places, and 1.40 times higher in school among students who had experienced SHS at home for four or more days in the last week. However, since the corrected variables are different from this study, care should be taken in comparing the two studies.

The biological mechanism by which SHS affects GAD has not been clearly reported. However, the expression of dopamine transporter mRNA was increased in the brains of laboratory rats continuously exposed to nicotine and SHS [26], and the GABAB (γ-aminobutyric acid type B) receptor involved in dopaminergic activity mRNA expression was likewise increased [27]. In other words, exposure to nicotine from SHS is related to the dopamine pathway in which neurotransmitters are activated, which subsequently induces depression and anxiety [28]. As such, SHS exposure can be a major biological factor that can have a detrimental effect on mental health, so it is necessary to confirm the causal relationship through more systematic and diverse experimental studies and cohort studies.

As a result of examining the demographic characteristics as sub-factors according to exposure to SHS, the group with exposure to SHS was relatively young at 19–34 years old, whereas the group without SHS exposure was similarly distributed across all age groups. This is presumed to be the result of exposure to SHS at home or at work, as the smoking rate of Koreans in their 40s is 24.0%, the highest among all age groups [1]. In addition, in terms of education level, the group with exposure to SHS had a high proportion of high school graduates, while the group without SHS exposure had a high proportion of graduates of college or higher education. According to Lee et al. [29], the geometric mean (geometric standard deviation) of the smokers’ urinary nicotine level was higher in the middle school graduate group than in the college graduate group, which was similar to the results of this study. Therefore, a more specific and systematic improvement policy for exposure to SHS should be made by identifying the home and work environment of subjects at high risk of SHS.

As a result of confirming oral-health-related factors as a sub-factor according to SHS exposure, the group without SHS exposure brushed their teeth more frequently after lunch and showed a higher percentage of oral examination and oral implants. Implants are a procedure to restore the lost function of teeth, and oral diseases can be prevented through regular oral examinations. As such, SHS exposure and oral health are highly correlated, and the study conducted on the elderly in Japan also reported that the number of remaining teeth decreased when subjects are exposed to SHS every day [9]. A domestic study also reported that the presence of gingival pain and bleeding symptoms was 1.201 times higher when there is SHS exposure compared to none [30]. In addition, exposure to SHS increases the risk of periodontal disease by 1.57 times, and it can cause periodontal disease as a result of local factors or systemic mechanisms as smoke flows into the mouth or nose through SHS [31]. Breivik T et al. [32] reported that when exposed to SHS, nicotine among tobacco components suppresses the immune response through nicotinic acetylcholine receptors, increasing the susceptibility to periodontal destruction and subsequently causing periodontal disease symptoms. Indirectly inhaling smoke alone causes systemic diseases identical to direct smoking, so blocking exposure to SHS should be treated as very important [33]. Therefore, it is necessary to recognize that not only direct smoking but also SHS is related to general health and oral health, and awareness improvement education to solve this problem will be important.

The limitation of this study involves the response bias that may have occurred in the process of writing self-response questions related to SHS [34,35]. In addition, the result value may be calculated differently depending on which confounding variable is included and adjusted. The effect of cumulative exposure to SHS could not be reflected because a detailed investigation related to the situation at the time of exposure, including the time of exposure to SHS, the size of the exposed place, and the ventilation of the place, was not conducted [36]. It is difficult to exclude the possibility that these limitations may bias the results of this study. In addition, although the results of a large-scale survey at the national level were analyzed, causality could not be confirmed as a cross-sectional and quantitative study. Despite these limitations, we can suggest the need to minimize exposure to SHS and practice oral health care in an effort to lower GAD since this study analyzed national data to confirm differences in SHS exposure, oral-health-related factors, and GAD among Korean adults. As a follow-up study, the risk of mental illness and oral disease is proposed according to the degree of secondhand smoking using national health examination data. Unlike schools and homes, there is no intermediary in public places where exposure to SHS is high, so government measures are required to recognize the dangers of SHS. In addition, there should be an anti-smoking education program and awareness education that can provide knowledge on SHS exposure prevention in the community integrated health promotion project and health policy implemented by local governments in Korea. In other words, it would be essential to prepare preventive policies through education to improve awareness of SHS exposure along with anti-smoking program policies. Lastly, this study is meaningful because it confirmed oral health management practice factors regarding the effect of SHS exposure in public places on GAD in adults. Therefore, it is necessary to expand anti-smoking education programs and promote awareness that exposure to SHS can cause harm to others.

## 5. Conclusions

This study analyzed data from the third year (2021) of the eighth National Health and Nutrition Examination Survey in Korea. The study subjects were adults who participated in the health survey, and a total of 3874 adults who answered if they were exposed to SHS in public places were selected as the final analysis subjects.

In conclusion, this study confirmed that adults’ exposure to SHS in public places is associated with GAD, and oral health care practice is an influencing factor in GAD. Therefore, it is necessary to inform the risk of exposure to SHS and manage oral health along with the implementation of preventive policies.

## Figures and Tables

**Figure 1 behavsci-13-00455-f001:**
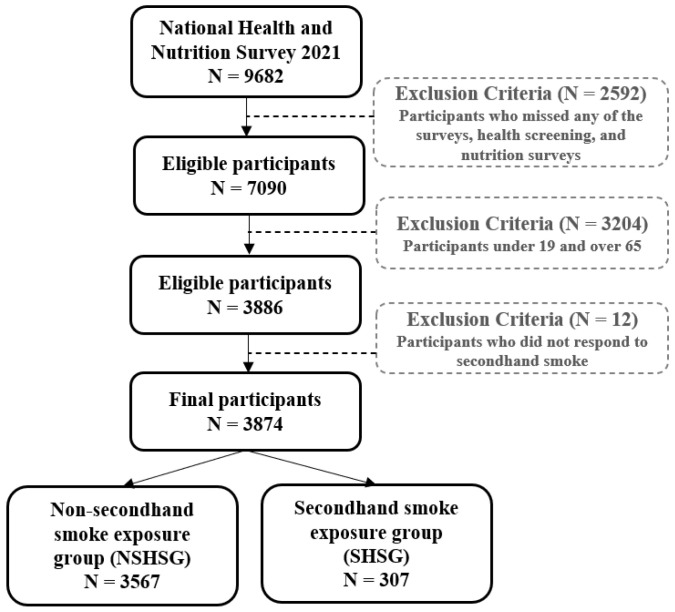
Study design.

**Table 1 behavsci-13-00455-t001:** Demographic characteristics of NSHSG and SHSG in public places (weighted %).

Characteristics	Division	SHSG (*N* = 307)	NSHSG (*N* = 3567)	*p* *
Gender	Male	163 (58.3)	1558 (50.7)	0.025
	Female	144 (41.7)	2009 (49.3)	
Age	19–34	113 (42.5)	838 (29.4)	<0.001
	35–44	56 (17.8)	767 (22.1)	
	45–54	83 (26.4)	904 (24.4)	
	55–64	55 (13.3)	1058 (24.1)	
Marital status	Married	180 (54.6)	2651 (69.8)	0.164
	Single	127 (45.4)	916 (30.2)	
Income level	5th quintile (highest)	97 (33.9)	1081 (30.5)	0.166
	4th quintile	84 (26.8)	971 (28.1)	
	3rd quintile	61 (19.3)	834 (23.7)	
	2nd quintile	39 (11.8)	454 (12.4)	
	1st quintile (lowest)	25 (8.2)	217 (5.3)	
Education level	<Elementary school	10 (6.0)	158 (12.4)	<0.001
	Middle School	13 (5.8)	241 (8.1)	
	High School	135 (45.5)	1347 (36.2)	
	>College	134 (42.7)	1671 (43.3)	

* By complex sample chi-square test, *p* < 0.05, SHSG; secondhand smoke group, NSHS; non-secondhand smoke group.

**Table 2 behavsci-13-00455-t002:** Difference in oral-health-status-related factors of NSHSG and SHSG in public places.

Characteristics	SHSG (*N* = 307)	NSHSG (*N* = 3567)	*p* *
Presence of existing natural teeth (yes, weighted %)	232 (98.7)	2480 (99.6)	0.084
Presence of maxillary prostheses (yes, weighted %)	33 (12.4)	508 (18.4)	0.060
Presence of mandibular prostheses (yes, weighted %)	37 (15.9)	585 (20.6)	0.180
Presence of maxillary anterior implants (yes, weighted %)	78 (23.2)	1121 (31.1)	0.012
Presence of maxillary posterior implants (yes, weighted %)	89 (25.8)	1367 (37.2)	0.001
Presence of mandibular anterior implants (yes, weighted %)	76 (22.4)	1099 (30.3)	0.010
Presence of mandibular molar implants (yes, weighted %)	99 (30.5)	1463 (39.4)	0.014
Complaints of chewing discomfort (yes, weighted %)	49 (14.1)	513 (13.2)	0.695
^†^ Self-recognized oral health status (M ± SD)	2.25 ± 0.05	2.10 ± 0.16	0.094
^†^ Chewing discomfort (M ± SD)	2.52 ± 0.04	2.60 ± 0.02	0.079
^†^ Speaking problems (M ± SD)	2.85 ± 0.03	2.88 ± 0.01	0.184

* By complex sample chi-square test, ^†^ by complex sample independent *t*-test, *p* < 0.05, SHSG; secondhand smoke group, NSHS; non-secondhand smoke group.

**Table 3 behavsci-13-00455-t003:** Differences in oral-health-behavior-related factors of NSHSG and SHSG in public place (Yes, weighted %).

Characteristics	Division	SHSG (*N* = 307)	NSHSG (*N* = 3567)	*t(p)* *
Tooth brushing	Brushing teeth before breakfast	139 (48.4)	1579 (44.0)	0.142
	Brushing teeth after breakfast	164 (50.0)	1986 (54.2)	0.240
	Brushing teeth before lunch	11 (4.0)	55 (1.7)	0.027
	Brushing teeth after lunch	153 (48.2)	1938 (54.2)	0.035
	Brushing teeth before dinner	12 (4.7)	97 (3.0)	0.194
	Brushing teeth after dinner	172 (53.4)	1989 (55.8)	0.512
	Brushing teeth after snack	19 (6.5)	160 (4.4)	0.147
	Brushing teeth before bed	188 (62.3)	1913 (54.1)	0.015
Dental treatment	Oral examination in the past year	142 (45.0)	1541 (41.7)	0.308
	Use a dental clinic	192 (63.3)	2161 (58.8)	0.180
	Oral examination treatment	177 (92.4)	2099 (96.4)	0.007
	Periodontal examination treatment	35 (17.6)	363 (16.0)	0.596
	Simple tooth decay treatment	54 (30.5)	534 (26.0)	0.209
	Dental pulp treatment	39 (21.4)	395 (18.5)	0.436
	Oral surgery treatment	38 (19.9)	310 (14.8)	0.077
	Oral preventive care	18 (8.2)	181 (8.7)	0.825

* By complex sample chi-square test, *p* < 0.05, SHSG; secondhand smoke group, NSHS; non-secondhand smoke group.

**Table 4 behavsci-13-00455-t004:** Differences in GAD-7 of NSHSG and SHSG in public place (M ± SD).

Characteristics	SHSG (*N* = 307)	NSHSG (*N* = 3567)	*p* *
GAD-1. I feel nervous, anxious, or irritable	0.48 ± 0.04	0.30 ± 0.01	<0.001
GAD-2. I can’t stop or control my worrying	0.49 ± 0.05	0.27 ± 0.01	<0.001
GAD-3. I worry too much about many things	0.82 ± 0.06	0.56 ± 0.02	<0.001
GAD-4. I find it difficult to be comfortable	0.40 ± 0.04	0.28 ± 0.01	0.009
GAD-5. I am so restless that I have a hard time staying still	0.25 ± 0.38	0.13 ± 0.01	0.002
GAD-6. I am easily annoyed or easily offended	0.64 ± 0.05	0.44 ± 0.02	<0.001
GAD-7. I feel afraid as if something terrible is about to happen	0.25 ± 0.04	0.12 ± 0.01	0.001
GAD-Total	3.32 ± 0.26	2.11 ± 0.07	<0.001

* By complex sample independent *t*-test, *p* < 0.05, SHSG; secondhand smoke group, NSHS; non-secondhand smoke group, GAD; generalized anxiety disorder.

**Table 5 behavsci-13-00455-t005:** The relationship between SHS exposure and GAD in public places.

Division	Model 1			Model 2			Model 3		
	*β*	*t*	*p* *	*β*	*t*	*p* *	*β*	*t*	*p* *
GAD-1	0.177	3.953	<0.001	0.191	4.253	<0.001	0.231	3.823	<0.001
GAD-2	0.226	4.174	<0.001	0.222	3.893	<0.001	0.270	3.653	<0.001
GAD-3	0.259	4.381	<0.001	0.248	3.945	<0.001	0.233	3.195	0.002
GAD-4	0.114	2.635	0.009	0.123	2.691	0.008	0.138	2.360	0.019
GAD-5	0.122	3.152	0.002	0.133	3.342	0.001	0.193	3.237	0.001
GAD-6	0.194	4.009	<0.001	0.210	3.898	<0.001	0.224	3.136	0.002
GAD-7	0.126	3.473	<0.001	0.124	3.417	0.001	0.133	2.671	0.008
Total	1.218	4.589	<0.001	1.255	4.520	<0.001	1.423	3.925	<0.001

* By complex sample linear regression analysis, *p* < 0.05, SHSG; secondhand smoke group, NSHS; non-secondhand smoke group, GAD; generalized anxiety disorder, Reference category; SHSG, Model 1 was unadjusted; Model 2 was adjusted for demographic characteristics; Model 3 was adjusted for demographic characteristics and oral-health-related factors, Model 1’s R^2^ = (1; 0.006, 2; 0.010, 3; 0.008, 4; 0.002, 5; 0.006, 6; 0.006, 7; 0.006, 8; 0.010), Model 2’s R^2^ = (1; 0.031, 2; 0.044, 3; 0.046, 4; 0.024, 5; 0.019, 6; 0.052, 7; 0.022, 8; 0.049), Model 3’s R^2^ = (1; 0.054, 2; 0.064, 3; 0.073, 4; 0.041, 5; 0.040, 6; 0.072, 7; 0.038, 8; 0.074).

## Data Availability

The data presented in this study are available on request from the corresponding author.

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
