# Peer review of "Differences in Oral Health and Generalized Anxiety Disorder According to Secondhand Smoke Exposure in Public Places"

_behavsci, 2023, doi:10.3390/bs13060455_

Round 1

Reviewer 1 Report

Key words: Smoking rate; secondhand smoking exposure; Health survey; GAD (generalized anxiety disorder) screening; public places; oral health status factors.   

In their 1. Introduction, the authors illustrate secondhand smoking (SHS) exposure in Korea, which has also prompted recent policies to prevent this matter on public health. Among the more acknowledged by literature health risks related to SHS, there are “dental loss” and “damage to mental health.” The authors have interest to a GAD (generalized anxiety disorder) screening on the topic, starting from the Korea National Health and Nutrition Examination Survey (KNHNES) of 2021. They set two simple hypotheses (H0 no association of SHS exposure to GAD/ H1 association) where H0 rejects H1 “to confirm the difference between secondhand smoking exposure in public places, the degree of oral health care, and GAD.” (See lines 66-67; hypotheses at lines 70-71). From this basic Model 1 of presence/lack of association, they further add a slightly modified Model 2 corrected by sociodemographic characteristics, and a Model 3 corrected by oral health related factors such as health management practices and pre-existing conditions (see lines 147-150).  

In 2. Materials and Methods, the authors highlight KNHNES is on 9,682 subjects, from which they have extracted 3,886 people aged from 19 to 65 years from the first year and third year of the 8th survey, and finally selected 3,874 subjects. Sampling procedure and exclusion criteria are shown in Fig. 1 at line 81. Sub-paragraph 2.4.1. Demographic Characteristics, shows the descriptive statistics categorization in the sample by authors. “Oral health status factors” previous to SHS in public spaces are divided in 7 different items, and “Oral health behaviors” in 2 items (i. brushing during the previous day, ii. not brushing during the previous day) which are then sub-divided into 8 items [a) before breakfast, b) after breakfast; c) before lunch, etc.]. 5 items, such as “gum disease treatment,” and “preventive treatment”) check instead the oral examination in the past year. “Oral health problems,” are exampled in “complaints of chewing discomfort,” and/or “speaking problems,” etc. (See lines 121-5).

7 questions with a 4-points Likert scale (0 ‘not disturbed at all,’ 1 ‘rather disturbed,’ 2 ‘mostly disturbed’, and 3 ‘disturbed every day’) have been introduced in the 3rd year of the 8th survey at KNHNE (Korea National Health and Nutrition Examination), and they measure properly GAD (so called, GAD-7). A summing procedure of scores shows a high/low level of GAD, respectively.

Sub-paragraph 2.9 Statistical Methods, illustrates the final sample of 3,874 people is divided into two groups: i. 307 people exposed to SHS in public spaces (SHSG), and ii. 3,567 people not exposed to SHS in public spaces (NSHSG). “Oral health status factors” previous to SHS in public spaces, “oral health behaviors,” and GAD-7 are used to conduct a linear regression analysis using a SPSS package by IBM is detailed at lines 167-174. Section 3. Results, shows some of the findings by authors, such as a “demographic confirmation of SHS exposure in public spaces by male subjects,” and the influence of the a) age range sub-classes are mentioned at line 104-105, as well of the b) marital status, c) level of education, or d) income level (always previously detailed at lines 106-9). Table 1 at line 188 shows the demographic characteristics of the two groups NSHSG and SHSG respectively, the authors have split their sample of 3,874 people. “Oral health status” in Korean adults change according to indoor SHS exposure in public spaces. There is also a difference in oral condition in the two sub-groups as shown in Table 2 at line 199. “Oral health behaviors,” now called by authors “Oral health management practices,” are also differentiated in the two sub-groups as shown in Table 3, line 211.

GAD-7 in Table 4 at line 221 expresses the most relevant differences according to indoor SHS exposure in public spaces or not. So that, the authors conclude for a significant association of indoor SHS exposure in public spaces with GAD. Table 5 at line 231 shows the effects of indoor SHS exposure in public spaces on GAD according to the three Models by authors, in particular 1) the  sign, 2) the true positive rate , and 3) the resulting p value with level of significance set at 0.05 (the desired risk of falsely rejecting H0). Section 4. Discussion and Section 5. Conclusions, praise the importance of GAD early detection for public health in Korea, because the association between SHS and mental health is not completely ascertained until now, nor its underlying biological mechanism “has been clearly reported.” The authors refer themselves to “the dopamine pathway in which neurotransmitters are activated, which subsequently induces depression and anxiety.” as referenced by Kim & Shim in 2019. The authors also mention other studies have reported damages to “Oral health factors” and/or “Oral health behaviors” because of indoor SHS exposure in public spaces. Biases of the research are highlighted in relation to the self-responding method at GAD-7 questions, and the difficulty in detecting latent variables such as “the situation at the time of SHS exposure,” and/or “the size of the exposed place.” For the rest, the study is self-evaluated as interesting because of the importance at expanding “anti-smoking education programs” and promoting awareness about SHS exposure (see lines 302-315).    

MINOR AMENDMENTS:

1.      The acronym SHS (secondhand smoke) must be solved also first time it appears in the main body at line 32. Acronyms must be solved each time they appear first in each part of the paper (abstract, text, figures, tables, appendices, etc.),

2.      The acronym GAD (generalized anxiety disorder) must be solved also first time it appears in the main body at line 57. Acronyms must be solved each time they appear first in each part of the paper (abstract, text, figures, tables, appendices, etc.),

3.      Please add the comma at the number 3874, line 84, as you did before at line 79,

4.      Please enlist the 7 different items of “oral health status factors” at lines 111-113: a) presence of existing natural teeth, b) maxillary prostheses, c) mandibular prostheses, etc.,

5.      Please delete one of the two “oral examination” at lines 119-120, because it is a repetition,

6.      Lines 140-2 are a repetition of lines 111-3, please delete them. Also, lines 142-145 are more or less a repetition of lines 115-124, think about to keep only one of this two descriptions (anyway repetition is less marked, and it can be kept if you prefer),

7.      Please think of the sentence at line 156: “The GAD is a three-point scale, with a…”. Since it ranges from 0 to 3, it should be a 4-points Likert scale (0,1,2,3),

8.      3874 at line 163: same at point 3 of this list,

9.      Please add the comma at the number 3567, line 188, Table 1, column NSHSG on the right,

10.   First column on the left at Table 1, line 188, please substitute “Marriage,” with “Marital status”,

11.   Second column on the left at Table 1, line 188, “Education level division,” please put all the levels with first letter of the first/second word in capital, or in small letter. Try to uniform the style in the paper of Tables, Figures, etc.

MAJOR AMENDMENTS:

12.   About the sample linear regression analysis, you mention at line 167-69, what about the “oral health problems,” are detected for example in “complaints of chewing discomfort,” and/or “speaking problems,” at lines 121-5? Is that not relevant to the linear regression? Please explain to a non-expert audience,

13.   Are the “oral health status factors” at lines 111-113 which are divided in 7 different items, such as: a) presence of existing natural teeth, b) maxillary prostheses, c) mandibular prostheses, etc., to be considered conditions are before SHS in public spaces? Please clarify this point to a non-expert audience,

14.   Since your data at Table 1, line 188, percentages in the two sub-groups are strongly weighted as you mention in the Table’s title. Can you please briefly explain how did you ponder the percentages? Please not also the last percentage at the bottom of the below chart on the left—"> College”—is inaccurate in your Table (please check and correct it). Same at Table 2: how did you ponder percentages?

Note: Well done, data in the sample are well-balanced.

15.   Since from your own findings “Oral health status” in Korean adults change according to indoor SHS exposure in public spaces—as you mention at lines 190-191—is that not another hypothesis to be mentioned at lines 70-71? A “change” could entail a “causation,” more than an “association” as correctly hypothesised in your H1, and as you know: «Correlation and causation are two concepts that are often used interchangeably but they are not the same thing. Correlation refers to the statistical relationship between two variables where a change in one variable is associated with a change in the other variable. However, this does not necessarily mean that one variable causes the other. On the other hand, causation refers to the relationship between two variables where one variable causes the other variable to change»,  

16.   If I were you, I would distribute better information in Sections 4. Discussions and 5. Conclusions, lessening the mentioning of others’ studies (they can be moved up shortly in the 1. Introduction) in Section 4 and highlighting the importance of SHS awareness in Section 5. 

References:

[1] Carreras, G.; Lugo, A.; Gallus, S.; Cortini, B. et alii Burden of disease attributable to second-hand smoke exposure: A systematic review. Prev Med. 2019 Dec, 129:105833. doi: 10.1016/j.ypmed.2019.105833. Epub 2019 Sep 7. PMID: 31505203.

[2] Flouris, A.D.; Koutedakis, Y. Immediate and short-term consequences of secondhand smoke exposure on the respiratory system. Curr Opin Pulm Med. 2011 Mar, 17(2):110-5. doi: 10.1097/MCP.0b013e328343165d. PMID: 21178628.

[3] Ksinan, A.J.; Sheng, Y.; Do, E.K.; Schechter, J.C. et alii Identifying the Best Questions for Rapid Screening of Secondhand Smoke Exposure Among Children. Nicotine & Tobacco Research 2021, 23(7), July, pp. 1217–1223, https://doi.org/10.1093/ntr/ntaa254

[4] Lin, L.Z.; Xu, S.L.; Wu, Q.Z.; Zhou, Y. et alii Exposure to second-hand smoke during early life and subsequent sleep problems in children: a population-based cross-sectional study. Environ Health. 2021 Dec 18, 20(1):127. doi: 10.1186/s12940-021-00793-0. PMID: 34920730; PMCID: PMC8684187.

[5] Mariano, L.C.; Warnakulasuriya, S.; Straif, K.; Monteiro, L. Secondhand smoke exposure and oral cancer risk: a systematic review and meta-analysis. http://dx.doi.org/10.1136/tobaccocontrol-2020-056393.

[6] Millar, W.J.; Locker, D. Smoking and oral health status. J Can Dent Assoc. 2007 Mar, 73(2):155. PMID: 17355806.

[7] Skipina, T.M.; Upadhya, B.; Soliman, E.Z. Secondhand Smoke Exposure is Associated with Prevalent Heart Failure: Longitudinal Examination of the National Health and Nutrition Examination Survey. Nicotine Tob Res. 2021 Aug 18, 23(9):1512-1517. doi: 10.1093/ntr/ntab047. PMID: 34213549.

[8] Tanaka, S.; Shinzawa, M.; Tokumasu, H.; Seto, K. et alii Secondhand smoke and incidence of dental caries in deciduous teeth among children in Japan: population based retrospective cohort study. BMJ 2015, 351:h5397 doi: 10.1136/bmj.h5397. Available from: https://www.bmj.com/content/bmj/351/bmj.h5397.full.pdf.  

[9] Ueno, M.; Ohara, S.; Sawada, N. et al. The association of active and secondhand smoking with oral health in adults: Japan public health center-based study. Tob. Induced Dis. 2015, 13, 19. https://doi.org/10.1186/s12971-015-0047-6.

Kind Regards,

Author Response

Comments and Suggestions for Authors

Thank you very much for your careful review.

We made corrections based on your comments.

We will try to do better research.

Sincerely

MINOR AMENDMENTS:

  1. The acronym SHS(secondhand smoke) must be solved also first time it appears in the main body at line 32. Acronyms must be solved each time they appear first in each part of the paper (abstract, text, figures, tables, appendices, etc.),

Answer: We have made corrections based on your comments.

  1. The acronym GAD(generalized anxiety disorder) must be solved also first time it appears in the main body at line 57. Acronyms must be solved each time they appear first in each part of the paper (abstract, text, figures, tables, appendices, etc.),

Answer: We have made corrections based on your comments.

  1. Please add the comma at the number 3874, line 84, as you did before at line 79.

Answer: We have made corrections based on your comments.

  1. Please enlist the 7 different items of “oral health status factors”at lines 111-113: a) presence of existing natural teeth, b) maxillary prostheses, c) mandibular prostheses, etc.,

Answer: We have made corrections based on your comments.

  1. Please delete one of the two “oral examination”at lines 119-120, because it is a repetition

Answer: We corrected the text.

  1. Lines 140-2 are a repetition of lines 111-3, please delete them. Also, lines 142-145 are more or less a repetition of lines 115-124, think about to keep only one of this two descriptions (anyway repetition is less marked, and it can be kept if you prefer),

Answer: Lines 140-142 were deleted, and the remaining sentences were not deleted because they were necessary.

  1. Please think of the sentence at line 156: “The GAD is a three-point scale, with a…”.Since it ranges from 0 to 3, it should be a 4-points Likert scale (0,1,2,3)

Answer: We corrected it on a 4-point scale. thank you

  1. 3874 at line 163: same at point 3 of this list,

Answer: We added a comma in the number. thank you

  1. Please add the comma at the number 3567, line 188, Table 1, column NSHSG on the right,

Answer: We added a comma in the number. thank you

  1. First column on the left at Table 1, line 188, please substitute “Marriage,” with “Marital status”,

Answer: We have made corrections based on your comments.

  1. Second column on the left at Table 1, line 188, “Education level division,” please put all the levels with first letter of the first/second word in capital, or in small letter. Try to uniform the style in the paper of Tables, Figures, etc.

Answer: We have corrected it in capital. thank you

MAJOR AMENDMENTS:

  1. About the sample linear regression analysis, you mention at line 167-69, what about the “oral health problems,”are detected for example in “complaints of chewing discomfort,” and/or “speaking problems,” at lines 121-5? Is that not relevant to the linear regression? Please explain to a non-expert audience,

Answer: We unified the variable names so that readers can easily understand them. Therefore, we also corrected the adjustments for the variables during the regression analysis.

  1. Are the “oral health status factors” at lines 111-113 which are divided in 7 different items, such as: a) presence of existing natural teeth, b) maxillary prostheses, c) mandibular prostheses, etc., to be considered conditions are before SHS in public spaces?Please clarify this point to a non-expert audience,

Answer: Oral health status is a variable to determine the difference between the two groups. In addition, since this study is a cross-sectional study, oral health status and SHS were confirmed at the time of the 2021 survey. Therefore, the relationship before and after SHS is unknown.

  1. Since your data at Table 1, line 188, percentages in the two sub-groups are strongly weighted as you mention in the Table’s title. Can you please briefly explain how did you ponder the percentages?Please not also the last percentage at the bottom of the below chart on the left—"> College”—is inaccurate in your Table (please check and correct it). Same at Table 2: how did you ponder percentages?

Note: Well done, data in the sample are well-balanced.

Answer: We have modified the number of [education level]. Percentages may not match because national data are analyzed by applying weights. And, Table 2 shows only those who answered [Yes].

  1. Since from your own findings “Oral health status” in Korean adults change according to indoor SHS exposure in public spaces—as you mention at lines 190-191—is that not another hypothesis to be mentioned at lines 70-71?A “change” could entail a “causation,” more than an “association” as correctly hypothesised in your H1, and as you know: «Correlation and causation are two concepts that are often used interchangeably but they are not the same thing. Correlation refers to the statistical relationship between two variables where a change in one variable is associated with a change in the other variable. However, this does not necessarily mean that one variable causes the other. On the other hand, causation refers to the relationship between two variables where one variable causes the other variable to change»,  

Answer: We modified our hypothesis.

  1. If I were you, I would distribute better information in Sections 4. Discussionsand 5. Conclusions, lessening the mentioning of others’ studies (they can be moved up shortly in the 1. Introduction) in Section 4 and highlighting the importance of SHS awareness in Section 5

Answer: We added about the importance of SHS awareness. thank you.

Reviewer 2 Report

This cross-sectional study supplements earlier results on Korean students by Shin et al. (ref. 15). Associations found between SHS and anxiety on the one hand and SHS and oral health on the other hand are no proof of causality. Generalization of results is not possible, because from 9,682 persons questioned only 3,874 were analyzed. One reason for exclusion was age <19 and >65 years. Surprising was that only 307 of them reported exposure to cigarette smoke by other people, but this question was restricted to "indoors in a public place (excluding designated smoking areas)" and is therefore worthless for assessment of exposure to SHS. The authors investigated only if persons -answering to this strange question with "yes"- also answered more frequently with "yes" to their 7 questions on anxiety. The result could support a hypothesis of the tobacco industry, that persons complaining about their exposure to indoor tobacco smoke in public places (where it is not allowed), are simply anxious, easily irritable and worried about many things, easily afraid, annoyed or offended, but not normal. The results do not contribute to numerous cohort and case-control studies which have proven damage of SHS to brain and behavior. The SHS group in this study is also younger than the comparison group, so that less frequent dental interventions are not surprising. Males prevailed (possibly because the authors did not care about women suffering from their husband's cigarettes at home). Model 2 and 3 are unclear and possibly unable to adjust for differences in education and socioeconomic status.

Author Response

Comments and Suggestions for Authors

This cross-sectional study supplements earlier results on Korean students by Shin et al. (ref. 15). Associations found between SHS and anxiety on the one hand and SHS and oral health on the other hand are no proof of causality. Generalization of results is not possible, because from 9,682 persons questioned only 3,874 were analyzed. One reason for exclusion was age <19 and >65 years. Surprising was that only 307 of them reported exposure to cigarette smoke by other people, but this question was restricted to "indoors in a public place (excluding designated smoking areas)" and is therefore worthless for assessment of exposure to SHS. The authors investigated only if persons -answering to this strange question with "yes"- also answered more frequently with "yes" to their 7 questions on anxiety. The result could support a hypothesis of the tobacco industry, that persons complaining about their exposure to indoor tobacco smoke in public places (where it is not allowed), are simply anxious, easily irritable and worried about many things, easily afraid, annoyed or offended, but not normal. The results do not contribute to numerous cohort and case-control studies which have proven damage of SHS to brain and behavior. The SHS group in this study is also younger than the comparison group, so that less frequent dental interventions are not surprising. Males prevailed (possibly because the authors did not care about women suffering from their husband's cigarettes at home). Model 2 and 3 are unclear and possibly unable to adjust for differences in education and socioeconomic status.

Answer: Thank you very much for your thoughtful review.

First of all, we modified the title meaning [Influence, effect] due to the limitation of cross-sectional study.

[Differences in Oral Health and Generalized Anxiety Disorder According to Secondhand Smoke Exposure in Public Places]

Also, we have put a lot of thought into what you are concerned about.

Therefore, we acknowledge the limitations of the part that can be investigated during the health checkup.

However, within the data given, we wanted to confirm the relevance between secondhand smoke in public places and oral health as well as anxiety.

Of course, exposure to secondhand smoke is very extensive.

There is a limit to confirming all of these, so we limited it to ‘public space’.

In addition, since public places [public institution buildings, schools, libraries, means of transportation, public sites, tourist accommodations, game providers, restaurants, cartoon rentals, etc.] are clearly specified in the national data questionnaire, we analyzed them using the relevant data.

We agree with your question.

However, we tried to utilize the given data, and since this is a limitation of the cross-sectional survey investigated at that time, we have sufficiently presented the contents of this as a limitation.

[The limitation of this study involves the response bias that may have occurred in the process of writing self-response questions related to SHS [35,36]. In addition, the result value may be calculated differently depending on which confounding variable is included and adjusted. The effect of cumulative exposure to SHS could not be reflected because a detailed investigation related to the situation at the time of exposure including the time of exposure to SHS, the size of the exposed place, and the ventilation of the place, was not conducted [37]. It is difficult to exclude the possibility that these limitations may bias the results of this study. In addition, although the results of a large-scale survey at the national level were analyzed, causality could not be confirmed as a cross-sectional and quantitative study.]

From your point of view, research conducted through surveys is not worth analyzing because it involves the subjective judgment of the respondents [Insincere answers, answers to consistent numbers, answers on behalf of others, etc.]

However, the research method should be diverse, and this study is also judged to be meaningful as a study that confirmed both the response results to the questionnaire and the examination results.

In addition, we proposed further research to supplement the limitations in [4 Discussion].

[As a follow-up study, the risk of mental illness and oral disease is proposed according to the degree of secondhand smoking using national health examination data.]

I ask for your consideration. Thank you.

Reviewer 3 Report

This study aims to provide basic data for minimizing exposure to environmental tobacco smoke (ETS) by examining its relationship with oral health behaviors. However, the authors found no significant association between exposure to ETS and oral health behaviors, which was expected, this finding should be highlighted in the abstract. Nonetheless, the article is well-structured and presents an important conclusion regarding the relationship between exposure to ETS and generalized anxiety disorder (GAD), which could have a significant impact on prevention efforts.

Author Response

Comments and Suggestions for Authors

This study aims to provide basic data for minimizing exposure to environmental tobacco smoke (ETS) by examining its relationship with oral health behaviors. However, the authors found no significant association between exposure to ETS and oral health behaviors, which was expected, this finding should be highlighted in the abstract. Nonetheless, the article is well-structured and presents an important conclusion regarding the relationship between exposure to ETS and generalized anxiety disorder (GAD), which could have a significant impact on prevention efforts.

Answer: Thank you for your review.

We edited the content of the abstract.

Round 2

Reviewer 1 Report

Dear authors,

I congratulate myself for this nice paper, which has been carefully amended. I noticed you have also amended calculations in Tables 1-3, and that's a strengthening argument for the goodness of research. The paper is improved, and I accept it for publication in Behavioural Sciences.

Best Wishes.